# The Multidirectional Effect of Azelastine Hydrochloride on Cervical Cancer Cells

**DOI:** 10.3390/ijms23115890

**Published:** 2022-05-24

**Authors:** Ewa Trybus, Teodora Król, Wojciech Trybus

**Affiliations:** Department of Medical Biology, The Jan Kochanowski University, Uniwersytecka 7, 25-406 Kielce, Poland; wojciech.trybus@ujk.edu.pl

**Keywords:** azelastine, oxidative stress, autophagy, apoptosis, mitotic catastrophe

## Abstract

A major cause of cancer cell resistance to chemotherapeutics is the blocking of apoptosis and induction of autophagy in the context of cell adaptation and survival. Therefore, new compounds are being sought, also among drugs that are commonly used in other therapies. Due to the involvement of histamine in the regulation of processes occurring during the development of many types of cancer, antihistamines are now receiving special attention. Our study concerned the identification of new mechanisms of action of azelastine hydrochloride, used in antiallergic treatment. The study was performed on HeLa cells treated with different concentrations of azelastine (15–90 µM). Cell cycle, level of autophagy (LC3 protein activity) and apoptosis (annexin V assay), activity of caspase 3/7, anti-apoptotic protein of Bcl-2 family, ROS concentration, measurement of mitochondrial membrane potential (Δψm), and level of phosphorylated H2A.X in response to DSB were evaluated by cytometric method. Cellular changes were also demonstrated at the level of transmission electron microscopy and optical and fluorescence microscopy. Lysosomal enzyme activities-cathepsin D and L and cell viability (MTT assay) were assessed spectrophotometrically. Results: Azelastine in concentrations of 15–25 µM induced degradation processes, vacuolization, increase in cathepsin D and L activity, and LC3 protein activation. By increasing ROS, it also caused DNA damage and blocked cells in the S phase of the cell cycle. At the concentrations of 45–90 µM, azelastine clearly promoted apoptosis by activation of caspase 3/7 and inactivation of Bcl-2 protein. Fragmentation of cell nucleus was confirmed by DAPI staining. Changes were also found in the endoplasmic reticulum and mitochondria, whose damage was confirmed by staining with rhodamine 123 and in the MTT test. Azelastine decreased the mitotic index and induced mitotic catastrophe. Studies demonstrated the multidirectional effects of azelastine on HeLa cells, including anti-proliferative, cytotoxic, autophagic, and apoptotic properties, which were the predominant mechanism of death. The revealed novel properties of azelastine may be practically used in anti-cancer therapy in the future.

## 1. Introduction

Drug resistance is a very big problem in most advanced cancers [1,2]. The biggest obstacle in cancer chemotherapy, including the treatment of cervical cancer, is resistance to cisplatin, among others, resulting from the induction of autophagy and inhibition of tumor cell apoptosis [3,4]. The process of programmed cell death can also be inhibited during oncogenesis. Cancer cells with multiple genetic and epigenetic alterations avoid apoptosis, which is initially triggered by the transformation process itself and then by the unfavorable tumor environment and the implemented therapy [5,6]. The resulting limitations in cancer therapy contribute to increased mortality. Therefore, recently, a new trend in worldwide research has become the search for alternative treatments, also inducing other types of cell death, especially among compounds already used in other therapies [5,7,8,9], an example of which may be antihistamines (AHs).

Antihistamines (AHs), due to their proven strong anti-inflammatory and anti-allergenic properties, are widely used worldwide as first-line drugs in the treatment of numerous allergic diseases [10]. Their mechanism of action involves stabilization of the inactive form of histamine H1 receptor, thereby blocking the action of histamine [11,12,13], which, as a major mediator of inflammatory response, not only underlies many allergic diseases [14], but is also directly involved in the regulation of biological processes during the development of various types of cancer, including cervical cancer [1,15,16]. Hence, in recent years, attention has been focused on the potential antitumor properties of antihistamines, both among the long-used and new second-generation representatives. Compounds have been identified, that alone or in combination with other drugs show significant activity against various types of cancer cells, confirmed both in vitro and in clinical trials. An example is astemizole, a second-generation drug, that has been described as an inhibitor of hepatocellular carcinoma proliferation [17] as well as an inducer of apoptotic death in various human melanoma cell lines [18,19]. In the case of terfenadine, the ability to induce apoptosis in prostate cancer cells has been demonstrated [20]. In turn, new representatives of AHs, more often used in practice, i.e., loratadine and its active metabolite desloratadine, improve survival in breast cancer [21,22,23] and skin melanoma [24]. Additionally, desloratadine has properties to induce apoptosis of T-cell lymphoma cells [25], and loratadine interferes with cell cycle progression of human colon cancer cells by increasing their sensitivity to radiation [26], and improves survival for ovarian cancer [7].

Azelastine hydrochloride (a phthalazinone derivative) is commonly used especially in the topical treatment of respiratory diseases, i.e., in allergic rhinitis (also in the course of asthma and COPD), vasomotor rhinitis, and as part of the prophylaxis and therapy of allergic conjunctivitis [27,28,29,30]. Furthermore, recent in vitro studies have demonstrated the ability of this compound to prevent and inhibit SARS-CoV-2 infection in nasal tissue [31]. Azelastine was also included in the list of compounds that exhibit lysosomotropic properties and have the ability to accumulate in the lungs when administered systemically, which creates the potential to achieve an effective drug concentration and was therefore recommended for use in patients with SARS-CoV-2 [32]. It should be emphasized that azelastine is a new representative of the second generation of H1 receptor antagonists, characterized by a different chemical structure than other preparations from this group [33] and high selectivity to the receptor, and thus low risk of side effects and very good tolerance both in adults and children [34,35,36,37,38]. It was also found that this group has an equivalent or faster onset of action compared to the first generation AHs [39]. Numerous scientific studies confirm that the biological properties of H1 receptor antagonists, including azelastine, also result from the possibility of non-receptor activity [13,40,41,42], which offers a broad perspective for the discovery of new therapeutic properties of these compounds.

In recent years, azelastine has also been tested for anti-inflammatory [43], antibacterial [44], and antiparasitic properties [45]. In turn, little attention has been paid to research into the potential anticancer mechanisms of this compound. So far, the property of azelastine to induce apoptosis in human colorectal adenocarcinoma cells (HT-29 line) has been described, where the tested compound at concentrations of 10µM-20µM, independently of the receptor, decreased the expression of Bcl-2 protein and caused significant changes in mitochondria [46]. In another study [47], azelastine at a concentration of 5µM sensitized KBV20C cells to the effects of vincristine (in combination administration), leading to decreased cell viability, arrest in G2 phase, and increased apoptosis. The results of the cited studies inspired the present study.

Therefore, due to the well-known resistance of HeLa cells to chemotherapy, which manifests itself by induction of autophagy and blockade of apoptosis, we decided to study the changes occurring in these cells under the influence of azelastine hydrochloride in the context of induction of apoptosis as well as other types of cell death as potential anticancer mechanisms of action of this compound.

## 2. Results

### 2.1. Azelastine Induces Apoptosis in HeLa Cells

Exposure of cells to azelastine resulted in an increase in the frequency of both cells in early (Annexin V-PE+/7-AAD−) and late apoptosis (Annexin V-PE+/7-AAD+).

At a concentration of 15 µM, apoptotic cells were over 26% (*p* ≤ 0.0001) and at 25 µM over 34% (*p* ≤ 0.0001) (Figure 1A,C). Azelastine at a concentration of 45 µM further increased the number of apoptotic cells to 60.13% (*p* ≤ 0.0001). Subsequent concentrations (60 µM and 90 µM) significantly increased the number of apoptotic cells, which were more than 93% and 98%, respectively (*p* ≤ 0.0001), with a clear predominance of cells with a late apoptotic phenotype.

Moreover, microscopic analysis (DAPI staining) showed that azelastine induces typical nuclear changes for apoptosis i.e., chromatin condensation and nuclear fragmentation, especially at concentrations of 60 µM and 90 µM (Figure 1E). The obtained results were dependent on the concentration of the test compound.

### 2.2. Azelastine Induces Caspase 3/7-Dependent Apoptosis

As shown in Figure 1B,C, caspase 3/7 activity increased significantly (*p* ≤ 0.0001) under the azelastine concentrations used. At 15 µM and 25 µM, cells with active caspase were more than 24% and 26%, respectively, and more than 55% at 45 µM concentration. The highest caspase 3/7 activity was shown for concentrations of 60 µM (96.4%) and 90 µM (98.7%). These results indicate a caspase 3/7 activation-dependent proapoptotic effect of azelastine.

### 2.3. Azelastine Inhibits the Viability of HeLa Cells

The MTT assay showed a highly statistically significant (*p* ≤ 0.0001) reduction in the ability of the cells to reduce the dye compared to the control, which was taken as 100% (Figure 1D). Already at the lowest concentration of 15 µM, the cell viability was 86% and at subsequent concentrations of 25 µM and 45 µM, it decreased significantly to 68.33% and 51.33%. However, the lowest percentage of living cells was obtained as a consequence of the highest concentrations of the test compound, i.e., 60 µM (8%) and 90 µM (about 4%). Azelastine inhibited mitochondrial metabolic activity to a concentration-dependent degree, which was also indicative of mitochondrial membrane damage.

### 2.4. Azelastine Generates ROS Inducing Changes in Mitochondrial Structure and Induces Endoplasmic Reticulum Stress

Compared to the control image (Figure 2(A1)), mitochondria with clear matrix and an irregular arrangement of mitochondrial cristae were observed already at the lowest azelastine concentration (15 µM) (Figure 2(A2)). As a result of 25 µM concentration, mitochondria showed a significant enlargement, a highly clear matrix, and a reduction in the mitochondrial cristae. Swollen channels of the rough endoplasmic reticulum were also visible in their close proximity (Figure 2A(3,3a)). In turn, the cytoplasm of cells exposed to 45 µM azelastine (Figure 2A(4,4a)) was dominated by swollen mitochondria with a strongly clear matrix, with disorganization of the inner mitochondrial membrane and pronounced damage to the cristae. Mitochondria were also characterized by disruption of the mitochondrial membrane, resulting in leakage of matrix contents into the cytoplasm (Figure 2(A4)). In contrast, visible in the microphotographs the rough endoplasmic reticulum appeared as dilated channels (Figure 2A(4,4a)). At subsequent azelastine concentrations of 60 µM and 90 µM (Figure 2A(5–6a)), the mitochondria were characterized by increased structure disorganization indicating significant damage, and in the vicinity of these structures remained the rough endoplasmic reticulum in the form of strongly widened and swollen cisterns. Compared with the control group, in which ROS (+) cells constituted 3.29%, treatment of HeLa cell with azelastine resulted in concentration-dependent intracellular ROS production (Figure 2C,D). The concentrations of 15 µM and 25 µM showed a small but statistically significant increase in ROS (+) cells to 12.4% and 24.8%, respectively (*p* ≤ 0.0001). Increasing the azelastine concentration to 45 µM resulted in increased generation of reactive oxygen species, ROS (+) cells accounted for more than 45% (*p* ≤ 0.0001). Significant levels of cellular ROS (+) were observed following azelastine treatment at concentrations of 60 µM (48.93%) and 90 µM (49.99%) (*p* ≤ 0.0001). The induction of reactive oxygen species generation correlated with a progressive decrease in mitochondrial membrane potential (Figure 2E,F).The lowest percentage of cells with mitochondrial membrane depolarization was shown at a concentration of 15 µM (9.76%) (*p* ≤ 0.0001). At 25 µM and 45 µM, cells with reduced mitochondrial membrane potential were 14.05% and 15.89%, respectively (*p* ≤ 0.0001). The highest percentage of cells with mitochondrial membrane depolarization (being more than 50%; *p* ≤ 0.0001) was found for concentrations of 60 µM and 90 µM. These results were confirmed in the imaging of rhodamine 123-labeled mitochondria, as it was shown that with increasing azelastine concentration, there is a gradual extinction of green fluorescence emission, significant in the range 45–90 µM (Figure 2B). The azelastine-induced increase in the level of reactive oxygen species, contributed to increased oxidative stress and stress of the rough endoplasmic reticulum and to the induction of apoptotic changes.

### 2.5. Azelastine Induces DNA Damage

The 48 h effect of azelastine induced a concentration-dependent increase in phosphorylated H2A.X in response to DNA double strand breaks (DSBs) (Figure 3B,C). The increase in DNA damage was as follows for the subsequent concentrations: 15µM (12.9%, *p* ≤ 0.0001), 25 µM (18.38%, *p* ≤ 0.0001), 45 µM (24.44%, *p* ≤ 0.0001), and 60 µM (28.92%, *p* ≤ 0.0001). At the highest concentration of 90 µM, cells with DBS accounted for more than 30% (*p* ≤ 0.0001) of all analyzed cells compared to the control (2.79%). DNA damage in azelastine-treated cells could have led to apoptosis.

### 2.6. Azelastine Inactivates the Bcl-2 Protein 

Azelastine induced inactivation of the anti-apoptotic protein Bcl-2 in HeLa cells (Figure 3A,D). At concentrations of 15 µM and 25 µM, cells with inactivated Bcl-2 protein represented 13.17% and 22.47% (*p* ≤ 0.0001), respectively, relative to the control (3.1%). Progressive changes in the expression of the test protein were shown at higher concentrations, i.e., 45 µM (40.56%), 60 µM (62.82%), and 90 µM (65%) (*p* ≤ 0.0001), which indicates a mechanism of proapoptotic action of azelastine involving the mitochondrial pathway.

### 2.7. Azelastine Enhances Vacuolization and Apoptotic Changes in HeLa Cells—Morphological Evaluation

In cells exposed to 48 h of azelastine, a significant concentration-dependent increase in the number of cells with vacuolization changes in the cytoplasm was observed (Figure 4D). Compared to control values (16 cells), the highest number of cells with enhanced vacuolization was observed at 15 µM (2119 cells) and 25 µM (2010 cells) (*p* ≤ 0.0001). Within the vacuole, a strong eosinophilic material destined for degradation was visible (Figure 4A(2,2a,3)). In contrast, at higher concentrations of the test compound, vacuolization changes showed a decreasing trend (Figure 4D). A lower but equally highly statistically significant result (1282 cells) was shown at a concentration of 45 µM (Figure 4A(4,4a)). On the other hand, at concentrations of 60 µM and 90 µM (Figure 4A(5–6a)), the presence of the lowest number of vacuolized cells was confirmed. 

The action of azelastine on tested cells resulted in the simultaneous appearance of cells with apoptotic changes such as reduced size, increased cytoplasm staining, pyknotic nucleus with chromatin condensation, and the presence of apoptotic bodies. At concentrations of 15 µM and 25 µM, apoptotic cells accounted for 236 and 963 (*p* ≤ 0.0001), respectively (Figure 4A(2–3a),D), compared to the control (11 cells). However, at 45 µM, the number increased significantly to 1708 (*p* ≤ 0.0001) (Figure 4A(4,4a),D). The highest value was observed at 60 µM (2988 cells) and 90 µM (2992 cells) (*p* ≤ 0.0001) (Figure 4A(5–6a),D). It should be noted that at a concentration of 45 µM (Figure 4A(4,4a)), there were cells with simultaneously observed features such as enhanced vacuolization of the cytoplasm and a pyknotic cell nucleus with partial chromatin condensation, indicating a gradual switch from vacuolization to apoptotic changes. The presence of phagocytosed apoptotic cells was observed within the cytoplasm of living cells (Figure 4A(4a,5a)) and those that were directed into the apoptosis pathway (Figure 4A(6,6a)), indicating induction of the efferocytosis process.

### 2.8. Azelastine Blocks Cells in S Phase and Reduces Mitotic Index

Cytometric analysis (Figure 4B,C) showed a statistically significant (*p* ≤ 0.0001) increase in the number of cells arrested in the S phase of the cell cycle, progressing with azelastine concentration. At the concentration of 15 µM, these cells accounted for 34.23%. Slightly higher results were obtained at a concentration of 25 µM (40.64%) and at 45 µM (44.47%). However, at 60 µM and 90 µM, there was a 2-fold increase in the number of cells in the above-mentioned phase as compared to the control (28.77%). At the same time, in the concentration range of 25–90 µM, there was a significant reduction in the number of cells in the G0/G1 phase (*p* ≤ 0.0001).

Comparison of cells incubated with azelastine at all concentrations used (15–90 µM) with cells from the control group (considered as 100%) showed statistically significant (*p* ≤ 0.0001) decrease in mitotic index (Figure 4E). Already at a concentration of 15 µM azelastine, the dividing capacity of the cells decreased significantly to 22% and this was also the highest recorded result, because at the other concentrations of the test compound, i.e., 25 µM, 45 µM, 60 µM and 90 µM, the mitotic index decreased to, respectively, 7%, 3%, 2%, and 1%. These changes demonstrate the antiproliferative properties of azelastine.

### 2.9. Azelastine Induces Mitotic Catastrophe

Morphological analysis showed that azelastine at 15 µM resulted in changes considered morphological markers of mitotic catastrophe (Figure 5). These included multiple abnormalities occurring during mitotic division, such as the presence of anaphase bridges (Figure 5(A2)), tripolar metaphase (Figure 5(A1)), and pentapolar anaphase (Figure 5(A3)). Azelastine also induced the formation of micronuclei (micronucleation) (Figure 5B), which were present in the highest and also statistically significant numbers at a concentration of 15 µM (Figure 5A(3–6)). Furthermore, the data obtained indicated clear multinucleation due to the action of the test compound (Figure 5B). The highest results were recorded at a concentration of 15 µM with 372 binucleated cells, 132 multinucleated cells, and 23 giant cells (at *p* ≤ 0.0001). At the next concentration of 25 µM, the results remained high in the range of statistically significant values, there were 267 binucleated cells, 90 multinucleated cells, and 12 giant cells found. However, at 45 µM azelastine, the number of binucleated, multinucleated, and giant cells significantly decreased to 47, 20, and 9, respectively, while at high concentrations of 60 µM and 90 µM, it was further reduced to levels below control values (Figure 5B).

Of note are the vacuolization (Figure 5C) and apoptotic (Figure 5D) changes observed simultaneously in cells with multinucleation. At low concentrations of azelastine (15 µM and 25 µM), vacuolization changes predominated over apoptotic ones, whereas at 45–90 µM, bi- and multinucleated cells were directed towards the apoptotic pathway. The results indicate that azelastine induces mitotic catastrophe, which precedes the onset of apoptosis.

### 2.10. Azelastine Enhances Degradation Processes

Analysis of changes at the ultrastructural level revealed numerous autophagic vacuoles in the cytoplasm of cells with azelastine at a concentration of 15 µM (Figure 6A(1–1b)); vacuoles were differentiated in size and content indicating different stages of degradation. In the studied cells, expanded Golgi apparatuses and dilated channels of the rough endoplasmic reticulum were present; these changes indicated the intensification of the process of synthesis of proteins crucial for subsequent stages of intracellular digestion. The presence of numerous mitochondria (Figure 6(A1a)) in the tested cells may result from the increased demand for ATP necessary for the macroautophagy process. Also at 25 µM concentration (Figure 6A(2–2b)), numerous and highly enlarged autophagic vacuoles containing material at different stages of degradation were shown, and vacuoles at the formation stage were also present (Figure 6(A2b)). In the lumen of these structures, large fragments of the cytosol with organelles were visible (Figure 6(A2,2b)), and some vacuoles took the form of emptiness and clearly demarcated from the cytoplasm spaces (Figure 6(A2a)). Swollen mitochondria (Figure 6(A2a)), dilated channels of rough endoplasmic reticulum, and reduced Golgi apparatus (Figure 6(A2)), whose membranes could be used for vacuole formation, were also observed in the cells. In contrast, in cells exposed to azelastine at 45 µM concentration (Figure 6A(3–3b)), the number of autophagic vacuoles was reduced; however, they had different shapes and covered a large area of the cytosol. In addition, altered mitochondria and single, slightly dilated channels of rough endoplasmic reticulum were present within the cytoplasm of these cells. When cells were treated with high concentrations of azelastine 60 µM and 90 µM (Figure 6A(4–5b)), the presence of secondary lysosomes was clearly marked alongside altered cell nuclei with local chromatin condensation (Figure 6(A4a), an expanded nuclear envelope (Figure 6A(4a,5,5a)), often with features of fragmentation (Figure 6(A5b)). There were also single, damaged mitochondria (Figure 6A(5a,5b)) and dilated channels of rough endoplasmic reticulum (Figure 6A(4–5a)). The demonstrated changes were dependent on the concentration of azelastine and indicated the intensification of the degradation processes. The progressive degradation observed at high concentrations may indicate a switch of cellular metabolism with the possibility of triggering programmed cell death.

### 2.11. Azelastine Activates Cathepsin D and L

As shown in the study, the effect of azelastine compared to control (which was taken as 100%) resulted in concentration-dependent changes in cathepsin D and L activity (*p* ≤ 0.0001) (Figure 6C). The highest increase in the activity of enzymes to 179.96% and 177.54% occurred at the concentrations of 15 µM and 25 µM, respectively. At 45 µM, the enzymes activity was found to be 173.89%. Further increase in the concentration of the test compound to 60 µM and 90 µM resulted in reduction of cathepsin D and L activities to 144.33% and 120.53%, respectively. The behavior of lysosomal enzymes is a reflection of the degradation processes activated by azelastine.

### 2.12. Azelastine Induces Autophagy by Increasing LC3 Protein Levels

According to the principle of the assay used, LC3 is a cytoplasmic protein involved in autophagosome formation during autophagy, which is translocated from the cytoplasm to the interior of autophagosomes, and its fluorescence is monitored cytometrically. According to the studies performed, azelastine induced autophagy depending on the concentration (Figure 6B). The highest intensity of fluorescence in cells was observed at concentrations of 15 µM and 25 µM, it was 139.3% and 143.36%, respectively, compared to the cells of the control group (gray area) (48.9%). With increasing azelastine concentrations, a gradual reduction in dye emission in labeled cells was observed to 95.3% at 45 µM and to 75.2% at 60 µM. At the highest concentration used (90 µM), a further reduction of the fluorescence intensity to 46.9% was demonstrated.

## 3. Discussion

Despite continuous advances in anticancer therapy, low treatment efficacy with concomitant high side effects is still a major problem [16]. Therefore, in the search for potential chemotherapeutic agents, particular attention is paid to the safety of the drug and its good tolerability [18]. Such features may be possessed by the well-studied new-generation H1 antihistamines, which have almost completely displaced the old-generation drugs used in anti-allergic treatment [48]. Another important aspect in the search for new oncological treatment options is the complexity of the oncological disease. The success of cancer therapy is also influenced by the possibility of modulating molecular and cellular factors found in the tumor and its microenvironment [16]. Thus, the identification of compounds with multidirectional mechanisms of action is crucial for the further development of anticancer therapies [6], and azelastine, used in anti-allergy treatment, may be such a drug.

The results obtained from our study allow us to conclude that the studied compound induced in HeLa cells two important processes for anticancer therapy, namely autophagy and apoptosis (Figure 4D). 

At low concentrations (15 µM and 25 µM), azelastine clearly promoted autophagy while apoptosis remained low. The induction of autophagy is indicated by an increased number of cells with intensified vacuolization of the cytoplasm (Figure 4A(2–3a),D). An important role in this process is played by the maintenance of an acidic pH inside the vacuole, which was documented by the presence of a strongly eosinophilic content within the large vacuoles of the studied cells (Figure 4A(2,3)). Adequate pH is necessary for the activity of lysosomal enzymes required to digest cellular material [49,50]. In our study, we showed that azelastine treatment caused a marked increase in the activity of lysosomal enzymes, i.e., cathepsin D and L (Figure 6C). The revealed concentration-dependent increase in lysosomal hydrolases activity was correlated with ultrastructural changes of studied cells, indicating an increase in degradative processes. The numerous autophagic vacuoles seen in the microphotographs (Figure 6A(1–2b)), which are very large and contain fragments of cytosol with organelles, indicate the presence of macroautophagy. This was also confirmed by examining the autophagy-specific marker, LC3 protein, where the highest fluorescence intensity (i.e., 139.3% and 143.36%) was found at the lowest concentrations of azelastine (15 µM and 25 µM, respectively) (Figure 6B).

As shown in Figure 4D, cells with morphological features typical of apoptosis clearly gained advantage at 45 µM azelastine, and with increasing concentrations of 60 µM and 90 µM, they constituted more than 90% of all analyzed cells. The switch of autophagy to apoptosis is documented in Figure 4A(4,4a), where cells with characteristics of both types of cell death are seen. Such a condition can be associated with progressive degradation of organelles, confirmed by the presence of giant autophagic vacuoles in cells loaded with 45 µM of azelastine (Figure 6A(3,3a)), as well as the presence of increased numbers of primary and secondary lysosomes at high concentrations (60 µM and 90 µM) of the test compound (Figure 6A(4–5b)). In the studied cells, enlarged mitochondria were visible next to the vacuoles (Figure 6A(5a,5b)), which according to the literature could be related to the increasing demand for ATP, necessary for enhanced autophagy as well as for triggering programmed cell death [51]. The nuclei of the cells also showed altered morphology, including chromatin condensation and fragmentation (Figure 6A(4a,5b)), which was confirmed by DAPI staining (Figure 1E). The pro-apoptotic effect was additionally confirmed by the cytometric method; it was shown that azelastine, depending on the concentration, significantly induced the number of apoptotic cells with the dominance of the late-apoptotic phenotype. These values increased as follows: up to 60% at a concentration of 45 µM, 93% at 60 µM, and 98% at 90 µM (Figure 1A,C).

Autophagy and apoptosis are interconnected and can occur in the same cell in response to a given stimulus simultaneously or separately [52,53]. According to Fimia and Piacentini [54], induction of apoptosis is often associated with increased autophagy. In the presence of apoptotic stimuli, autophagy may be an adaptive response or a distinct type of cell death [55]. The tendency to change the regulation of both processes demonstrated in our studies was dependent on the concentration of azelastine. The targeting of cells to the apoptotic pathway was likely the result of a failed attempt to restore cellular homeostasis as a consequence of increased cellular stress [56]. During excessive autophagy, mitochondria tend to show accelerated production of reactive oxygen species due to increased oxidative phosphorylation. A slight, but statistically significant increase in the ROS level (Figure 2C,D), was noted already at the lowest concentrations of the tested compound (15 µM and 25 µM). The ROS values increased significantly at the concentration of 45 µM, which could have triggered the apoptosis process in the tested cells. Similar results were obtained by the team of Nicolau-Galmés [55] in a study on terfenadine, an old generation antihistamine, which enhanced autophagy and consequently led to the induction of apoptosis. 

The oxidative stress activated by azelastine in HeLa cells was correlated with the simultaneous increase in the level of phosphorylated H2A.X (Figure 3B,C). The results obtained in this study indicate the participation of ROS in inducing DNA damage, which could have been a signal to trigger apoptosis. The significantly reduced division capacity of HeLa cells (Figure 4E) and their arrest in the S phase of the cell cycle (Figure 4B,C) may be associated with the DNA damage response. As shown in the literature, cell proliferation may be crucial for tumor development and progression, and histamine may be the main mediator of this process in various types of cancer [16]. On the other hand, DNA damage and inhibition of cell proliferation are among the important mechanisms of action of anticancer drugs [57]. The various properties of azelastine demonstrated in this study can also be used in anticancer therapy. 

The antiproliferative properties of azelastine are also confirmed by the mitotic catastrophe induction capacity demonstrated in studies, documented in Figure 5A showing the abnormal course of mitosis. This process was most likely induced by DNA damage, and resulted in demonstrated changes such as multinucleation and micronucleation (Figure 5A(1–6,B)). Of particular note is the presence of giant, multinucleated cells (Figure 5(A4)). At high concentrations of the tested compound, cells with significant nuclear changes eventually underwent apoptosis (Figure 5D), which is considered one of the necessary final steps in the course of mitotic catastrophe. The mitotic catastrophe shows a strong mechanistic relationship with the cellular and molecular changes accompanying carcinogenesis and therefore seems to be a preferentially stimulated process in cancer cells [58,59,60]. Compounds promoting mitotic death, such as azelastine, may be a promising therapeutic alternative for apoptosis-resistant cancer cells.

In cells, the functions of “stress sensors” are performed by mitochondria and they are the central executors of apoptosis [61] as well as the course of mitotic catastrophe [59]. As our results showed, the induction of apoptosis by azelastine was also associated with the mitochondrial pathway. At the level of submicroscopic studies, already at low concentrations of azelastine, mitochondria were enlarged with a clear matrix (Figure 2A(2,3,3a)). However, under the influence of high concentrations, enhanced changes were demonstrated, with significant mitochondrial damage (Figure 2A(4–6a)) and disorganization of the inner membrane. At the same time, the cytometric analysis determined the highest percentage of cells with depolarization of the mitochondrial membrane (over 50%) for the concentrations of 60 µM and 90 µM (Figure 2E,F). The violation of the mitochondrial membrane integrity was confirmed by the concentration-dependent gradual quenching of green fluorescence emissions from labeled mitochondria (Figure 2B), which was also associated with the demonstrated inactivation of the Bcl-2 protein (Figure 3A,D) and activation of the executive caspases (Figure 1B,C). We also demonstrated the cytotoxic effect of azelastine related to the reduction of metabolic activity of mitochondria using the MTT test. Depending on the concentration, this compound reduced the viability of HeLa cells (Figure 1D), which at the highest concentration (90 µM) was only 4%. According to the studies by Cornet-Masana [62] conducted on leukemia lines, mitochondria in cancer cells are characterized by numerous changes, which, according to Pathania’s team [63], makes them more susceptible to therapies aimed at the metabolism of cancer cells.

The analysis performed at the level of submicroscopic changes revealed that mitochondrial disorganization is also accompanied by significant changes in the profile of the rough endoplasmic reticulum. It has been reported that even at low concentrations of azelastine, there is significant dilatation of reticulum channels (Figure 2A(3,3a)). These changes intensified as a consequence of the action of increasing concentrations of the tested compound, and at its high concentrations, they became significantly swollen (Figure 2A(5–6a)), which can be explained by the stress of the reticulum. The revealed changes in the endoplasmic reticulum homeostasis may be induced by an increased level of ROS [64,65], which is also confirmed by the results of our research.

Cao and Kaufman [66] emphasize in their works the importance of spatial and functional distribution in cells of organelles such as mitochondria and endoplasmic reticulum. Also in the analyzed electronograms, close proximity of altered mitochondria and expanded channels of rough endoplasmic reticulum was demonstrated (Figure 2A(5a–6a)), indicating a functional relationship between these organelles and which may be relevant to the processes regulating apoptotic cell death. Pro-apoptotic factors derived from mitochondria induce signals from the rough endoplasmic reticulum, which in turn cause changes in the mitochondria. On the other hand, reticulum stress can lead to mitochondrial dysfunction and consequent oxidative stress, followed by impaired homeostasis and apoptosis [67,68,69]. Apoptosis involving endoplasmic reticulum stress has attracted a lot of attention in recent years [64]. Mild stress of cancer cells can lead to the activation of adaptive mechanisms, however, therapeutic benefits of compounds that induce endoplasmic reticulum stress and put cells on the apoptosis pathway have been confirmed for certain types of cancer cells [70,71]. In our studies, azelastine induced in HeLa cells oxidative stress, stress of rough endoplasmic reticulum, and mitochondrial dysfunction, which by reinforcing each other, disrupted cellular functions and activated proapoptotic signals [65,66,68]. A similar mechanism of action has been reported for terfenadine, an old generation antihistamine in relation to A375, HT144, and Hs294T cell lines [55]. However, in studies on the action of rupatadine, ebastine, and loratadine in relation to acute myeloblastic leukemia cells, the cytotoxicity of these compounds consisted of bidirectional, mitochondrial-lysosomal action, ROS generation, and reduction of mitochondrial metabolic activity, which led to the activation of caspase 3 and 7 and induction of the apoptosis pathway [62]. 

Efferocytosis-phagocytosis of dead cells was also observed under the action of azelastine (Figure 4A(4a,5,6,6a)). According to the literature, this process under certain conditions can be performed by “non-professional phagocytes” [72,73,74]. In the context of carcinogenesis, efferocytosis suppresses the body’s natural immune response, then facilitates the immune escape of tumor cells while promoting the tumor microenvironment [50]. This process not only affects the proliferation, invasion, metastasis, and angiogenesis of cancer cells, but also regulates adaptive responses and decreases the positive response to radiotherapy and to many commonly used anticancer antibodies and chemotherapeutic agents [75]. The data obtained in our study indicate, that initially azelastine at low concentrations (Figure 4(A4a)) induced efferocytosis in the context of an adaptive response of HeLa cells, and then cells with phagocytosed apoptotic cells were directed to the apoptotic pathway (Figure 4A(6,6a)). Such action is in contrast to that of traditional therapies, which induce apoptosis of tumor cells and increase subsequent efferocytosis, which suppresses the inflammatory response [76]. Thus, the demonstrated property of azelastine indicates an additional possibility of the interference of the compound in the tumorigenesis, and at the same time, fits in the current view of combining traditional therapies with therapies targeting the efferocytosis process in order to improve their effectiveness [50].

## 4. Material and Methods

### 4.1. In Vitro Culture Conditions

Human cervical adenocarcinoma cells (HeLa cells) were cultured in a Direct Heat incubator (Thermo Scientific, Waltham, MA, USA), under standard culture conditions, i.e., 37 °C and 5% CO_2_, on a modified DMEM medium (GIBCO, New York, NY, USA), containing 10% fetal calf serum (Biowest, Nuaillé, France) and a mixture of antibiotics (penicillin G, streptomycin, amphotericin B) (Corning, Manassas, VA, USA). The HeLa cells were purchased from the American Type Tissue Culture Collection (Rockville, MD, USA). Cells were treated for 48 h with azelastine hydrochloride (≥98% HPLC), (4-[(4-chlorophenyl)methyl]-2-(1-methylazepan-4-yl) phthalazin-1-one hydrochloride), which was purchased from Sigma Aldrich (St. Louis, MO, USA). The following concentrations of the test compound were used in the experiment: 15 µM, 25 µM, 45 µM, 60 µM, and 90 µM. 

According to the literature data, the tested concentrations are used in research on antihistamine drugs conducted on cancer cell lines. Control cells were cultured in complete maintenance medium without the addition of the test compound.

### 4.2. Assessment of Cell Viability—MTT Test

The level of cytotoxicity of azelastine against HeLa cells was determined using MTT (3-(4,5-dimethyl-2-yl)-2,5-diphenyltetrazolium bromide) reduction assay. Cells seeded in Falcon 96-well plates (Fisher Scientific, Waltham, MA, USA) after azelastine treatment were stained with MTT solution (1 mg/mL) (Sigma Aldrich, St. Louis, MO, USA). After 2 h of incubation of the cells with the dye, dimethylsulfoxide (DMSO) was applied to solubilize the formed formazan crystals. Optical density was measured at 570 nm using a Synergy 2 multi-detector microplate reader (BioTek, Winooski, VT, USA). Cell viability was calculated in comparison with the control group using Gen5 software.

### 4.3. Visualization of Apoptotic Cells under A Fluorescence Microscope

Morphological evaluation of nuclei of control and tested cells was performed using 4′,6-diamidino-2-phenylindole (DAPI) staining. Cells cultured in dishes (Falcon, Fisher Scientific, Waltham, MA, USA) were stained with 2.5 µg/mL DAPI solution (Sigma Aldrich, St. Louis, MO, USA) for 15 min and then washed with PBS. The preparations were analyzed using a Nikon Eclipse Ti epi-fluorescence inverted microscope (Nikon Instruments Inc., Melville, NY, USA).

### 4.4. Detection of Apoptosis

Phosphatidylserine externalization in azelastine-exposed cells was assessed using Annexin V and Dead Cell test kit (Merck Millipore, Burlington, MA, USA). Control and azelastine treated cells were detached using 0.25% trypsin-EDTA (Corning, New York, NY, USA), centrifuged and washed with PBS. Then, cells were stained with annexin V (100 µL) for 20 min at room temperature in the dark. The fluorescence intensity was analyzed using a Muse analyzer (Merc Millipore, Burlington, MA, USA).

### 4.5. Activity of Caspase-3/7

The level of caspase-3/7 activation was measured using a caspase-3/7 assay kit (Merck-Millipore, Burlington, MA, USA). After 48 h of incubation with azelastine, cells were harvested by trypsinization and incubated at 37 °C with 5 µL of Caspase-3/7 working solution (as per protocol). Then, 150 µL of Caspase 7-AAD working solution was added to the cells. Detection of caspase-positive cells was performed using a Muse analyzer (Merck-Millipore, Burlington, MA, USA).

### 4.6. Analysis of Ultrastructural Changes

Cells for electron microscopy were fixed in 3% glutaraldehyde (Serva Electrophoresis GmbH, Heidelberg, Germany) followed by 2% OsO4 (Spi, West Chester, PA, USA) in cacodyl buffer. The material was then dehydrated in an ascending series of ethanol solutions (10–99.8%) and embedded in Epon 812 epoxy resin (Serva Electrophoresis GmbH, Heidelberg, Germany), followed by polymerization at 40 °C and 60 °C. The epoxy blocks were cut into ultra-thin sections on a Leica EM UC7 ultramicrotome (Leica Biosystems, Wetzlar, Germany), and the obtained preparations were further contrasted with uranyl acetate and lead citrate. Analysis was performed using a Tecnai G2 Spirit transmission electron microscope (FEI Company, Hillsboro, OR, USA) equipped with a Morada camera (Olympus, Soft Imagine Solutions, Münster, Germany). The interpretation of the changes in azelastine-exposed cells was based on the image of control cells.

### 4.7. Measurement of the Mitochondrial Membrane Potential (Δψm)

The decrease in Δψm was analyzed using the Muse MitoPotential Assay kit (Merck Millipore). Cells after incubation with azelastine were resuspended in 95 µL of Muse MitoPotential working solution and incubated at 37 °C for 20 min. The cells were then stained with 7-AAD dead cell marker (5 µL) at room temperature for 5 min, and the cell suspension was analyzed by flow cytometry.

### 4.8. Microscopic Evaluation of Changes in the Potential of Mitochondrial Membrane

After 48 h incubation with azelastine, cells were fixed in 4% paraformaldehyde and then incubated for 30 min with rhodamine 123 (Sigma Aldrich, St. Louis, MO, USA) at a concentration of 5 µg/mL ethanol. The used fluorochrome binds to metabolically active mitochondria, so the fading of fluorescence is proportional to the decrease in mitochondrial membrane potential. The cells were then washed with PBS and analyzed under a Nikon A1 confocal microscope based on a Nikon Eclipse Ti inverted microscope (Nikon Instruments Inc., Melville, NY, USA) and equipped with Nikon Nis Elements AR software (Nikon Instruments Inc., Melville, NY, USA).

### 4.9. Oxidative Stress Analysis

The Muse Oxidative Stress Assay kit (Merck Millipore, Burlington, MA, USA) based on intracellular detection of superoxide radicals was used to investigate the level of reactive oxygen species. As according to the manufacturer’s instructions, cells were treated with Muse Oxidative Stress Reagent working solution (190 µL) after 48 h incubation with azelastine. Samples were then incubated at 37 °C for 30 min and the percentage of gated ROS (−) and ROS (+) cells with ROS activity were analyzed.

### 4.10. Assessment of Bcl-2 Protein Phosphorylation

Changes in Bcl-2 phosphorylation in HeLa cells were assessed using the Muse™ Bcl-2 Activation Dual Detection Assay kit (Merck-Millipore, Guyancourt, France) according to the manufacturer’s instructions. Two direct conjugated antibodies were used in the kit, i.e., phospho-specific anti-phospho-Bcl-2 (Ser70)-Alexa Fluor^®^ 555 and a conjugated anti-Bcl-2-PECy5 antibody to measure total Bcl-2 expression levels. The degree of activation of the Bcl-2 pathway was assessed by measuring Bcl-2 phosphorylation relative to total Bcl-2 expression in the tested cells.

### 4.11. DNA Damage Assessment

To determine whether azelastine causes DNA damage, cells were fixed and permeabilized with Muse Fixation Buffer and Permeabilization Buffer reagents, followed by staining with anti-phospho-Histone H2A.X (Ser139) and anti-phospho-ATM (Ser1981) antibodies according to the instructions for the Muse H2A.X Activation Dual Detection kit (Millipore, Darmstadt, Germany).

### 4.12. Cell Cycle Analysis

Cells were analyzed using the Muse Cell Cycle Assay Kit (Merck Millipore, Burlington, MA, USA). Cells were trypsinized and centrifuged, and the obtained cell pellet was fixed in 70% ice-cold ethanol. Cells were then treated with Muse Cell Cycle Reagent (Merck Millipore, USA) for 30 min and then analyzed with a Muse analyzer (Merck Millipore, Burlington, MA, USA).

### 4.13. Visualization of Morphological Changes and Assessment of the Dividing Capacity of HeLa Cells

Cells were cultured on sterile coverslips in Falcon dishes (Fisher Scientific, Waltham, MA, USA) in DMEM medium supplemented with azelastine (test cells) or without test compound (control cells). Methanol-fixed cells were stained with Harris hematoxylin and eosin, then dehydrated in an ascending series of ethanol solutions and immersed in xylene. Each preparation was analyzed based on a control image, taking into account changes mainly concerning cell nucleus (presence of bi- and multinucleated cells, giant cells, cells with micronuclei, with chromatin condensation, with pyknotic nucleus), cytoplasm (increased or decreased pigmentation, vacuolization changes, presence of apoptotic bodies), and mitotic division (presence of cells in particular phases of division and abnormal mitotic figures). Quantitative and qualitative analysis of morphological changes in the studied cells and photographic documentation were performed using a Nikon Eclipse 80i microscope with Nikon NIS Elements D 3.10 software (Nikon Instruments Inc., Melville, NY, USA). The mitotic index was evaluated by determining the number of cells in each phase of mitotic division, and the result was expressed as a percentage. In preparations, 3000 cells each were analyzed in three independent experiments (9000 cells/concentration), and the final score for a given trait was the mean value.

### 4.14. Evaluation of Cathepsin D and L Activity Levels

After 48 h of incubation with azelastine, cells were trypsinized, resuspended in 0.25 M sucrose solution and homogenized using a Potter S homogenizer (Sartorius, Gottingen, Germany). The homogenate was initially centrifuged at an overload of 700× *g*, for 10 min. The extranuclear supernatant was then centrifuged at 20,000× *g* overload for 20 min, and the obtained lysosomal pellet was resuspended in Triton X-100 (Sigma-Aldrich, St. Louis, MO, USA). The activities of degradative enzymes, cathepsin D and L, were determined in the lysosomal fraction according to the modified Langner’s method. According to the procedure, 2% azocasein (Sigma-Aldrich, St. Louis, MO, USA), 0.2 M acetate buffer pH = 5.0, and 10% TCA (+4 °C) were used. After incubation at 37 °C, samples were centrifuged, and enzyme activity was measured by colorimetric method at 366 nm using a Spekol 1500 spectrophotometer (Analytik Jena GmbH, Jena, Germany). Simultaneously, the total protein content (at 680 nm) was determined using the Lowry’s method modified by Kirschke and Wiederanders. Enzyme activity was expressed as μmol/mg protein/hour.

### 4.15. LC3-Antibody Detection

The level of azelastine-induced autophagy was assessed by cytometric assay using Autophagy LC3 antibody (Merck Millipore, Burlington, MA, USA). The kit includes reagent to selective membrane permeabilization (Autophagy Reagent A) that allows to distinguish between cytosolic and autophagic LC3. This is accomplished by extracting the cytosolic protein while protecting the LC3 that is translocated to and remains intact in autophagosomes. Addition of Anti-LC3 Alexa Fluor^®^ 555 and Autophagy Reagent B to the cells allows quantification of LC3 by measuring fluorescence using flow cytometry. According to the protocol, cells were seeded in Falcon 96-well plates. Autophagy A reagent in EBSS medium (Corning, Corning, NY, USA) was then added to the cells and incubated for 4 h under CO_2_ atmosphere, followed by washing with HBSS (Corning, Corning, NY, USA), trypsinization, and centrifugation. The supernatant was removed and anti-LC3 Alexa Fluor^®^ 555 and Autophagy Reagent B were added to the cells and incubated on ice for 30 min in the dark. The samples were then analyzed using flow cytometry technique. Cells that were treated with serum-free medium for 4 h were used as a positive control.

### 4.16. Statistical Analysis

Statistical analysis of the study results was performed using one-way analysis of variance (ANOVA) with multiple post-hoc comparisons using Tukey’s test. Differences were considered statistically significant at *p* < 0.05. Statistica 10.0 software (StatSoft, Krakow, Poland) was used for data analysis.

## 5. Conclusions

In our study we demonstrated potential anticancer properties of azelastine based on autophagic, proapoptotic, cytotoxic, or antiproliferative activity, which, taking into account safety of its application and potent anti-inflammatory properties, can be regarded as features of a compound that is part of the current canon of fight against cancer. Azelastine may be therefore an alternative method of oncological treatment, which requires further research.

## Figures and Tables

**Figure 1 ijms-23-05890-f001:**
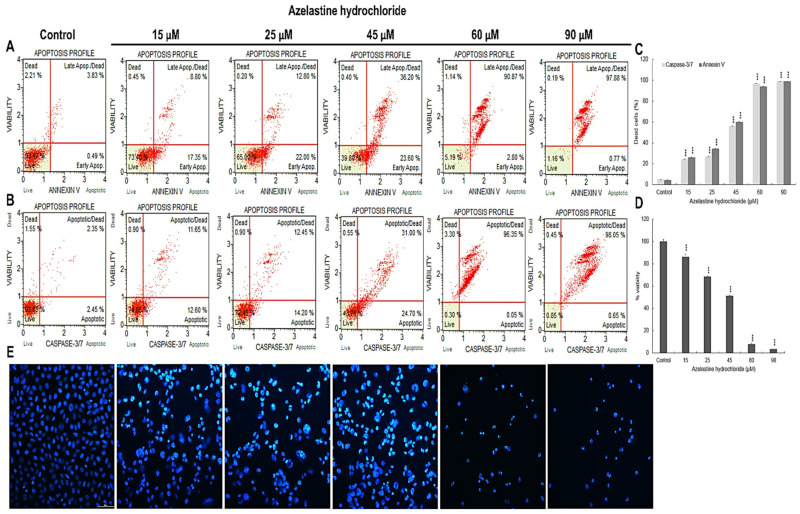
Proapoptotic effects of azelastine hydrochloride. Control cells and treated for 48 h with azelastine at concentrations of 15 µM, 25 µM, 45 µM, 60 µM, and 90 µM. (**A**) Level of apoptosis determined by annexin V-PE/ 7-AAD staining. Live cells (annexin V-PE−/7-AAD−), cells in early (annexin V-PE+/7-AAD−) and late-stage apoptosis (annexin V-PE+/7-AAD+), and necrotic cells (V-PE−/7-AAD+). (**B**) Changes in 3/7 caspase activity. Live cells (caspase 3/7-/7-AAD−), cells in early (caspase 3/7+/7-AAD−) and late apoptosis (caspase 3/7+/7-AAD+), dead cells (caspase 3/7-/7-AAD+). (**C**) Percentage of apoptotic cells dependent on azelastine concentration. (**D**) Cell viability as determined by the MTT assay. (**E**) Changes in nuclei of cells labeled with 4′,6-diamidino-2-phenylindole (DAPI). Control cells showing normal cell nuclei morphology. Cells treated with azelastine showing changes typical of apoptosis, i.e., marked condensation of chromatin and fragmentation of cell nucleus. Images were taken at 4000× magnification. Data representative of three parallel experiments correspond to mean values ± standard error (SE). Differences were statistically confirmed at: *** *p* < 0.001.

**Figure 2 ijms-23-05890-f002:**
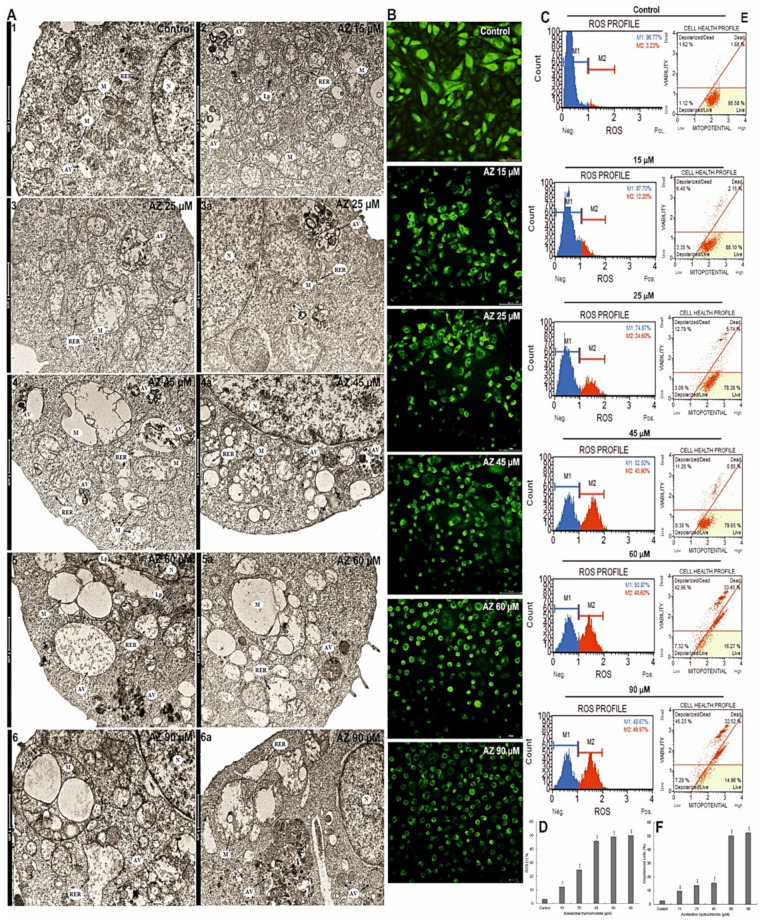
Changes in mitochondria, induction of oxidative stress, and endoplasmic reticulum stress in HeLa cells caused by the action of azelastine hydrochloride. Control cells (**A1**). Azelastine concentration-dependent ultrastructural changes indicative of apoptosis (**A2**–**6a**); brightening of the matrix and irregular arrangement of cristae in the mitochondria of cells subjected to the 15 µM concentration (**2**); at 25 µM, visible enlarged mitochondria with reduction of mitochondrial cristae remaining in close proximity to the dilated channels of the rough endoplasmic reticulum (**3**,**3a**); at a concentration of 45 µM, mitochondria with enhanced damage characteristics are present, i.e., strongly enlarged with disruption of the mitochondrial membrane (**4**) and damaged mitochondrial cristae (**4a**) and altered rough endoplasmic reticulum in the form of dilated channels (**4**,**4a**); at concentrations of 60 µM (**5**,**5a**) and 90 µM (**6**,**6a**), visible mitochondria with severe disorganization of the structure indicating damage, and rough endoplasmic reticulum located in their vicinity with strongly enlarged and swollen cisterns. Explanation of abbreviations: N—nucleus, M—mitochondria, AG—Golgi apparatus, RER—rough endoplasmic reticulum, AV—autophagic vacuoles, Lp—primary lysosomes, Ls—secondary lysosomes. Images were taken at 11,500× magnification. (**B**) Gradual and azelastine concentration-dependent loss of green fluorescence derived from rhodamine 123-labeled mitochondria. (**C**) Generation of reactive oxygen species and (**D**) percentage of ROS (+) cells as a result of azelastine. (**E**) Changes of mitochondrial membrane potential (Δψm) and the percentage of cells with mitochondrial membrane depolarization (**F**) at different azelastine concentrations. Each sample was analyzed in triplicate. The differences were statistically confirmed at: *** *p* < 0.001.

**Figure 3 ijms-23-05890-f003:**
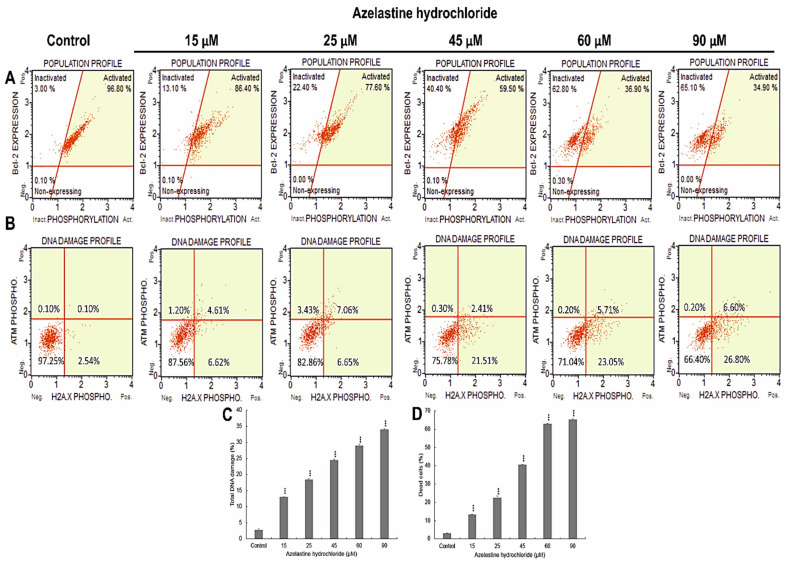
Bcl-2 protein inactivation and DNA damage demonstrated in cells exposed to 48 h action of azelastine hydrochloride. (**A**) Cells expressing Bcl-2 are clustered in the top two quadrants of the dot plot (inactivated and activated). Over 60% are dephosphorylated after treatment with azelastine at 60 µM and 90 µM, confirming inactivation of the Bcl-2 signaling pathway. (**B**) Azelastine at concentrations of 15–90 µM generates DNA damage that induces DNA repair mechanisms such as ɣH2AX. (**C**) Azelastine concentration-dependent percentage of cells with double-stranded DNA damage and (**D**) percentage of cells with Bcl-2 protein inactivation. Data representative of three parallel experiments correspond to mean values ± standard error (SE). Differences were statistically confirmed at: *** *p* < 0.001.

**Figure 4 ijms-23-05890-f004:**
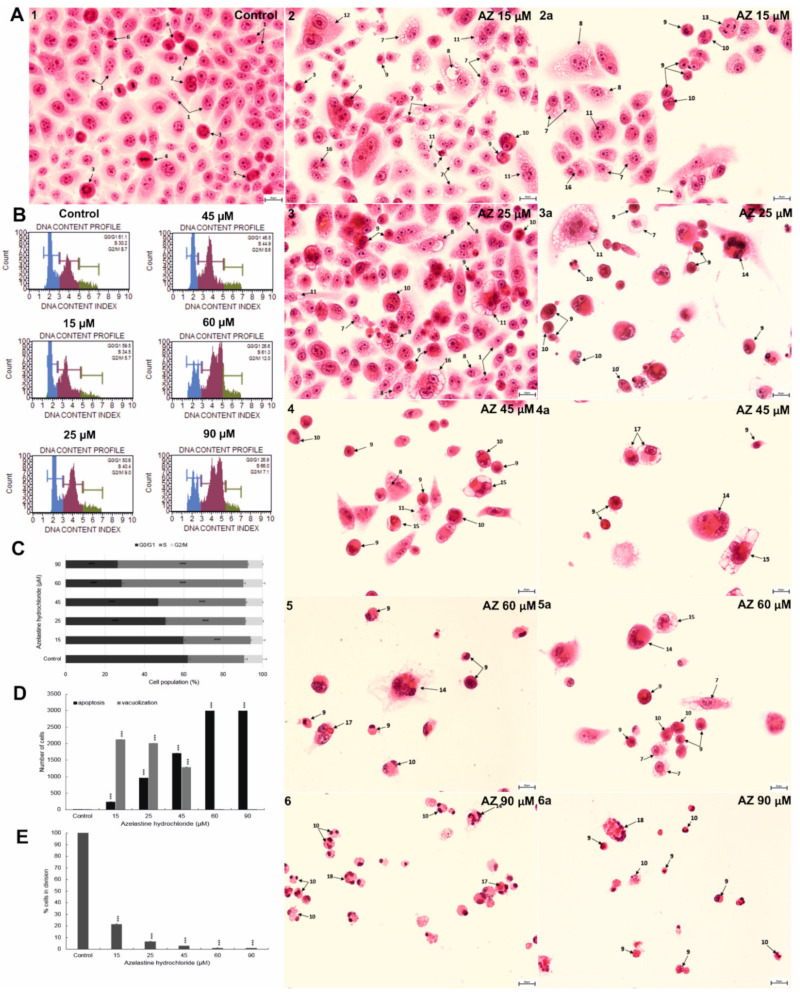
Morphological changes indicating the induction of vacuolating and apoptotic changes and a decrease in the dividing capacity of Hela cells as a consequence of 48 h treatment with azelastine hydrochloride. (**A1**) Control cells with normal morphology, including interphase cells and cells with multiple mitotic figures. (**A2**–**6a**) Cells treated with azelastine at concentrations of 15–90 µM; (**A2**–**3a**) cells with numerous vacuolization changes in the cytoplasm, strongly eosinophilic material is visible within the vacuoles, which is destined for degradation, indicating the process of autophagy; (**A4**,**4a**) cells with intensive cytoplasmic vacuolization and a pyknotic cell nucleus with visible partial chromatin condensation; (**5**–**6a**) strong pro-apoptotic effect of azelastine at concentrations of 60 µM and 90 µM, expressed by the presence of numerous apoptotic cells and cells with efferocytosis. Explanation of markings: 1—interphase, 2—prophase, 3—prometaphase, 4—metaphase, 5—telophase, 6—cytokinesis, 7—vacuolization of cytoplasm, 8—vacuoles with visible digestion material, 9—apoptotic cells, 10—binucleated cells in apoptosis, 1—-binucleated cells with vacuolization, 12—giant cells, 13—abnormal segregation of chromosomes, 14—multinucleated cells in apoptosis, 15—cells with features of vacuolization and apoptosis, 16—multinucleated cells with vacuolization, 17—efferocytosis, 18—cells with phagocytosed material (by efferocytosis), which were directed toward the apoptotic pathway. Hematoxylin and eosin staining. Images were taken at 4000× magnification. (**B**) Cell cycle changes of HeLa line cells treated for 48 h with azelastine at concentrations of 15–90 µM analyzed by flow cytometry. (**C**) Percentage of cells in different phases of the cell cycle indicating blocking of cells in S phase. (**D**) Azelastine concentration-dependent number of vacuolated and apoptotic cells; at concentrations of 15–25 µM, azelastine induced vacuolization changes; at a concentration of 45 µM, there was a reduction in vacuolization changes in favor of apoptotic cells, while at concentrations of 60–90 µM, azelastine promoted apoptosis. (**E**) Changes in the mitotic index indicating an inhibition of the dividing capacity of azelastine. Average values from three independent experiments. The differences were statistically confirmed at: *** *p* < 0.001.

**Figure 5 ijms-23-05890-f005:**
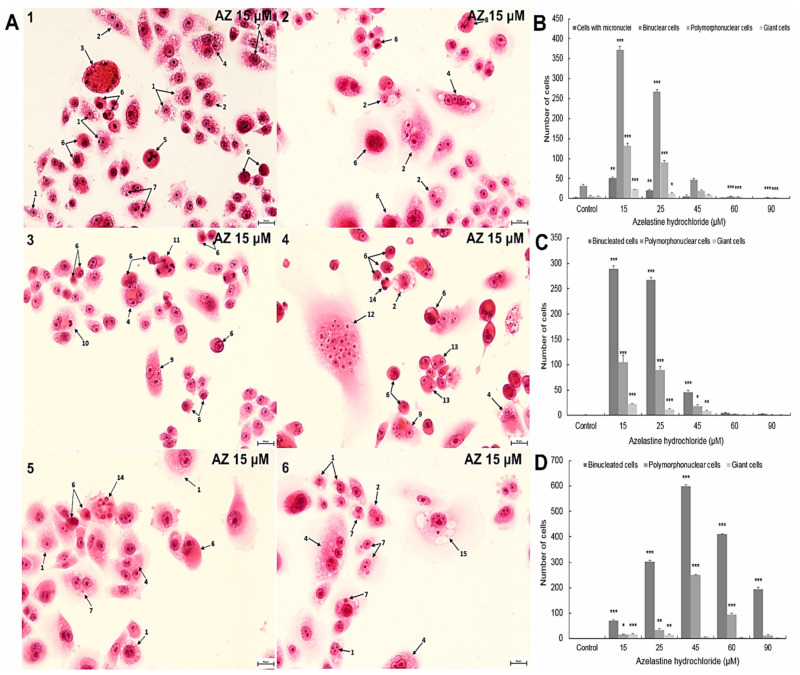
Morphological markers of mitotic catastrophe induced in HeLa cells exposed to 48 h treatment of azelastine hydrochloride. (**A**) Intensified changes (multipolar mitosis, micronucleation, and multinucleation) in cells treated with 15 µM azelastine. Explanation of markings: 1—cells with vacuolization, 2—binucleated cells with vacuolization, 3—apoptotic giant cell, 4—multinucleated cells with vacuolization, 5—tripolar metaphase, 6—apoptotic cells, 7—cells with micronuclei, 8—anaphase bridges, 9—multinucleated cells with micronuclei, 10—efferocytosis, 11—pentapolar anaphase, 12—giant cells, 13—binucleated cells in apoptosis, 14—multinucleated cells in apoptosis, 15—giant cells with vacuolization and micronuclei. Hematoxylin and eosin staining. Images were taken at a magnification of 4000×. (**B**) Distribution of cells with micronuclei, bi-, multinucleated cells, and giant cells at different concentrations of azelastine 15–90 µM. Azelastine concentration-dependent induction of vacuolization (**C**) and apoptotic (**D**) changes in bi-, multinucleated, and giant cells. Average values from three independent experiments. The differences were statistically confirmed at: * *p* < 0.05, ** *p* < 0.01, *** *p* < 0.001.

**Figure 6 ijms-23-05890-f006:**
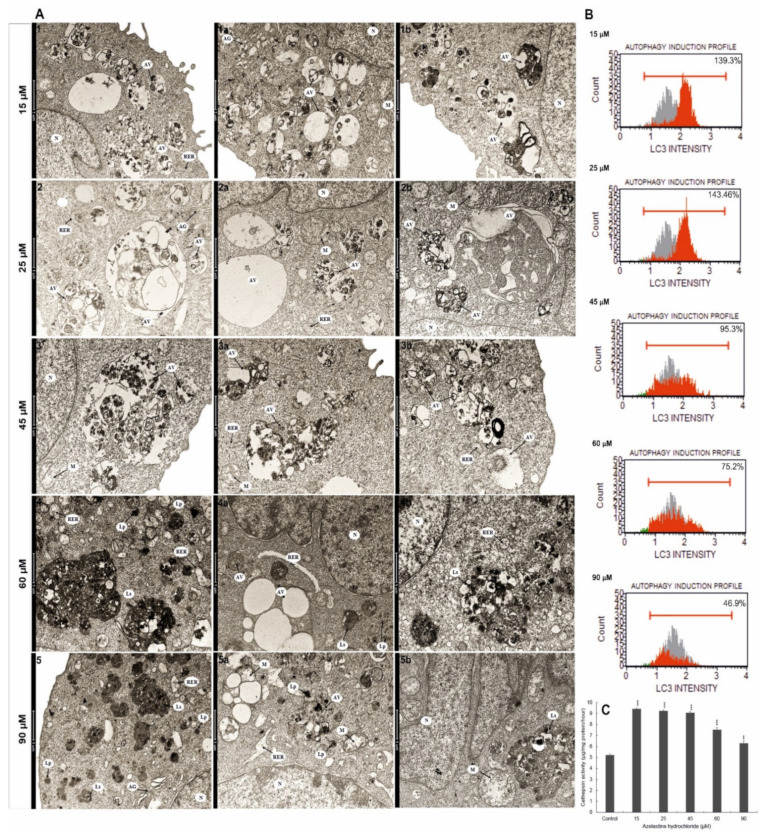
The intensification of degradation processes in HeLa cells with azelastine hydrochloride. (**A**) Ultrastructural changes; (**1**–**1b**) in the cytoplasm of cells very numerous autophagic vacuoles filled with material at various stages of degradation can be seen, and an extensive Golgi apparatus and dilated channels of the rough endoplasmic reticulum; (**2**–**2b**) clear changes in cells indicating a progressive degradation process, i.e., multiple vacuoles at different stages of digestion; (**3**–**3b**), the image shows vacuoles differentiated in terms of shape, covering large areas of the cytosol, indicating intensive degradation; (**4**–**5b**) dominant changes in the form of numerous primary and secondary lysosomes and changes at the level of the cell nucleus (local chromatin condensation, enlarged nuclear envelope and fragmentation). The changes obtained in the concentrations of 60 µM and 90 µM of azelastine indicate a progressive degradation as well as the initiation of cell death by apoptosis. (**B**) Histograms of cells indicating azelastine-induced autophagy as manifested by an increase in fluorochrome fluorescence intensity (red area) indicating LC3 protein activation. In the concentration range of 45–90 µM, the fluorescence intensity decreased, which argued for a switch of autophagy to apoptosis. Cells were stained with anti-LC3/Alexa Fluor^®^ 555 conjugated antibody and the fluorescence intensity was measured cytometrically. (**C**) Changes in cathepsin D and L activities (mean ± SE) in the lysosomal fraction of HeLa cells after 48 h of exposure to different concentrations of azelastine. Results are the average of three independent experiments. Differences were statistically confirmed at: * *p* < 0.05, ** *p* < 0.01, *** *p* < 0.001.

## Data Availability

The data that support the findings of this study are available from the corresponding author upon reasonable request.

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
