# Peer review of "The Multidirectional Effect of Azelastine Hydrochloride on Cervical Cancer Cells"

_ijms, 2022, doi:10.3390/ijms23115890_

Round 1

Reviewer 1 Report

The authors have selected a known anti-histamine drug Azelastine to evaluate its anti-cancer properties in one of the aggressive cancers - cervical cancel. 

However, the rationale for choosing an anti-histamine to evaluate as anti-cancer agent is not explained. Why do authors think that anti-histamine may have anti-cancer properties? any prior evidence? and why cervical cancer? No mechanism of action is explained. Azelastine causes apoptosis at very high concentrations (540-60uM). There is no evidence to show that this is not a off-target effect. Is this cell killing selective to cancer cells? No normal tissue control is used. Drugs at this high in vitro concentration is not only difficult to achieve in vivo, but will induce lot of off -target side effects. High concentrations of drug causes cancer cell death - evidence of apoptosis does not explain how BCL2 is inhibited and why.

Therefore, this manuscript as written with very broad conclusions that are not supported by the data presented is not ready for publication. 

Author Response

Responses to the review 1

The choice of the cell line for this study was dictated by the fact that in the case of cervical cancer cells, the biggest obstacle to chemotherapy is resistance to cisplatin, which is due to the induction of autophagy by these cells and at the same time inhibition of apoptosis (Xia et al. 2016). The aim of our research was to demonstrate the changes occuring in HeLa cells under the influence of azelastine in the context of inducing the process of apoptosis as well as other types of cell death.

The mechanism of action of antihistamines involves stabilization of the inactive form of the histamine H1 receptor, thereby blocking the action of histaminÄ™, which, as a major mediator of inflammatory response, not only underlies many allergic diseases, but is also directly involved in the regulation of biological processes during the development of various types of cancer, including cervical cancer (Massari, et al. 2018; Zhang et al. 2020). According to previous reports, compounds with mechanisms of action effective in anticancer therapy have been identified among the second-generation LPH (the information is presented in detail in the introduction), which indicates the possibility of using azelastine hydrochloride in this area.

The compound used in the research - azelastine hydrochloride, is a new generation antihistamine, characterized by high receptor selectivity, and at the same time a limited risk of side effects and very good tolerance in both adults and children. It is widely used as a first-line drug in the treatment of allergic diseases due to its strong antiallergic effect. As proven in numerous studies, azelastine also shows a mechanism of action without the participation of the receptor, which in turn gives the prospect of discovering new properties and the possibility of therapeutic use of this compound. In recent years, azelastine has also been tested for its anti-inflammatory, antibacterial, antiviral and antiparasitic properties. In turn, little attention has been paid to research into the potential anticancer mechanisms of this compound. So far, the property of azelastine to induce apoptosis in human colorectal adenocarcinoma cells (HT-29 line) has been described, where the tested compound at concentrations of 10µM-20µM, independently of the receptor, decreased the expression of Bcl-2 protein and caused significant changes in mitochondria (Hu et al., 2021). In another study (Kim et at. 2019), azelastine at a concentration of 5 µM sensitized KBV20C cells to the effects of vincristine (in combination administration), leading to decreased cell viability, arrest in G2 phase, and increased apoptosis. The results of the cited studies inspired the present study.

The compound concentrations used in our study are used in antihistamine studies conducted on cancer cell lines (Liang et al., 2020; Matsumoto et al. 2021; Soule et al. 2010).

The reduction in Bcl-2 protein expression shown in our research may be related, inter alia, to in violation of the integrity of the mitochondrial membrane documented on microphotographs showing changes in the ultrastructure of cells after exposure to azelastine, with quenching of green fluorescence emissions derived from rhodamine 123-labeled mitochondria, and with a decrease in mitochondrial membrane potential (Δψm) demonstrated by cytometric analysis.

Thank you very much for your review.

Reference work:

Xia J, Yu X, Song X, Li G, Mao X, Zhang Y. Inhibiting the cytoplasmic location of HMGB1 reverses cisplatin resistance in human cervical cancer cells. Mol Med Rep. 2017 Jan;15(1):488-494. doi: 10.3892/mmr.2016.6003. Epub 2016 Dec 7. PMID: 27959427.

Zhang E, Zhang Y, Fan Z, Cheng L, Han S, Che H. Apigenin Inhibits Histamine-Induced Cervical Cancer Tumor Growth by Regulating Estrogen Receptor Expression. Molecules. 2020 Apr 23;25(8):1960. doi: 10.3390/molecules25081960. PMID: 32340124; PMCID: PMC7221565.

Massari NA, Nicoud MB, Medina VA. Histamine receptors and cancer pharmacology: an update. Br J Pharmacol. 2020 Feb;177(3):516-538. doi: 10.1111/bph.14535. Epub 2018 Dec 13. PMID: 30414378; PMCID: PMC7012953.

Hu HF, Xu WW, Li YJ, He Y, Zhang WX, Liao L, Zhang QH, Han L, Yin XF, Zhao XX, Pan YL, Li B, He QY. Anti-allergic drug azelastine suppresses colon tumorigenesis by directly targeting ARF1 to inhibit IQGAP1-ERK-Drp1-mediated mitochondrial fission. Theranostics. 2021 Jan 1;11(4):1828-1844. doi: 10.7150/thno.48698. PMID: 33408784; PMCID: PMC7778598.

Kim JY., Kim KS., Kim IS, Yoon S. Histamine Receptor Antagonists, Loratadine and Azelastine, Sensitize P-gp-overexpressing Antimitotic Drug-resistant KBV20C Cells Through Different Molecular Mechanisms. Anticancer Research. 2019 Jul;39(7):3767-3775. DOI: 10.21873/anticanres.13525. PMID: 31262903.

Liang YC, Chang CC, Sheu MT, Lin SY, Chung C, Teng CT, Suk FM. The Antihistamine Deptropine Induces Hepatoma Cell Death through Blocking Autophagosome-Lysosome Fusion. Cancers (Basel). 2020 Jun 18;12(6):1610. doi: 10.3390/cancers12061610. PMID: 32570749; PMCID: PMC7352610.

Matsumoto N, Ebihara M., Oishi S., Fujimoto Y., Okada T., Imamura T. Histamine H1 receptor antagonists selectively kill cisplatin-resistant human cancer cells. Sci Rep. 2021 Jan 15;11(1):1492. doi: 10.1038/s41598-021-81077-y. PMID: 33452347; PMCID: PMC7810706.

Soule BP, Simone NL, DeGraff WG, Choudhuri R, Cook JA, Mitchell JB. Loratadine dysregulates cell cycle progression and enhances the effect of radiation in human tumor cell lines. Radiat Oncol. 2010 Feb 3;5:8. doi: 10.1186/1748-717X-5-8. PMID: 20128919; PMCID: PMC2829588.

Reviewer 2 Report

In the current study, the authors demonstrated the multidirectional effects of azelastine on HeLa cells, including anti-proliferative, cytotoxic, autophagic and apoptotic properties, which reveals potential anticancer properties

It is an interesting and well-conducted study. The introduction fully described the state of the art and the material and methods are clear. In the manuscript, the results are clearly described and convincing. 

I suggest acceptabling for publication after minor point:

1) In Figure 1: C. Cell viability as determined by the MTT assay and D. Percentage of apoptotic cells dependent on azelastine concentration.  Are reversed?

2) the following title needs to be corrected “2.2. Azelastine-induced caspase-dependent apoptosis 3/7”

3) the authors should improve the resolution of the figures in particular of all histograms

4) in line 365: “The decreased level of the LC3 protein at high concentrations (60 μM and 90 μM) of azelastine indicates high cytotoxicity of the tested compound with the possibility of autophagy switching into apoptosis”. I think this statement does not correlate with the title (2.12. Azelastine induces autophagy by increasing LC3 protein levels).  

Author Response

Responses to the review 2

As suggested, the following changes have been included in the submitted manuscript:

Ad.1. The descriptions of panels C and D in figure 1 have been changed accordingly in the legend: "Percentage of apoptotic cells dependent on azelastine concentration" is marked as C, while „Cell viability determined by MTT test” is marked as D.

Ad.2. In line 115: the title „2.2. Azelastine-induced caspase-dependent apoptosis 3/7” has been corrected to „2.2. Azelastine induces caspase 3/7-dependent apoptosis”.

Ad.3. The resolution of the figures was improved.

Ad.4. In line 365: the sentence „Decreased LC3 protein levels at high concentrations (60 μM and 90 μM) of azelastine indicate high cytotoxicity of the test compound with the possibility of autophagy transitioning to apoptosis" was removed. This statement is in the Discussion.

Thank you very much for the sent reviews and all comments, which contributed to the improvement of the quality of the article.

Reviewer 3 Report

The manuscript “The multidirectional effect of azelastine hydrochloride on cervical cancer cells” is very interesting and well-written. Data was well-presented. There are some minor problems though.

  1. the font size in Figures 1a, b, c, d is too small to read. Likewise, it would be great to make the font size bigger in other figures, especially the quantification bar plot.

Author Response

Responses to the review 3

As suggested, the font size in the figures and the English language was corrected in the submitted manuscript.

Thank you very much for the sent reviews and all comments, which contributed to the improvement of the quality of the article.